# Detection and Monitoring of Topography Changes at the Tottori Sand Dune Using UAV-LiDAR

**DOI:** 10.3390/s26010302

**Published:** 2026-01-02

**Authors:** Jiaqi Liu, Jing Wu, Soichiro Okida, Reiji Kimura, Mingyuan Du, Yan Li

**Affiliations:** 1Institute of Environmental Systems Science, Shimane University, 1060 Nishikawatsu-cho, Matsue, Shimane 690-8504, Japan; 2Graduate School of Sciences and Technology for Innovation, Yamaguchi University, 1677-1 Yoshida, Yamaguchi 753-8511, Japan; wu.jing@yamaguchi-u.ac.jp; 3Arid Land Research Center, Tottori University, 1390 Hamasaka, Tottori 680-0001, Japan; okida@tottori-u.ac.jp (S.O.); rkimura@tottori-u.ac.jp (R.K.); 4Xinjiang Institute of Ecology and Geography, Chinese Academy of Sciences, 818 South Beijing Road, Urumqi 830011, China; dumy@ms.xjb.ac.cn; 5College of Engineering, Northeast Agricultural University, NO.600 Changjiang Street Xiangfang District, Harbin 150000, China; liyanneau@neau.edu.cn

**Keywords:** UAV-LiDAR sensors, environmental sensing, GNSS-RTK, ground control points (GCPs), coastal dune monitoring

## Abstract

**Highlights:**

**What are the main findings?**
An integrated UAV-LiDAR sensor system with GNSS-RT and optimized ground control point (GCP) design enables centimeter-level digital elevation models (DEMs) from LiDAR point clouds, which are validated against GNSS reference data.Subtle dune dynamics, including elevation changes of up to 0.4 m associated with wind-driven sand transport, were detected.

**What are the implications of the main findings?**
This study demonstrates UAV-LiDAR as a robust environmental sensing framework for coastal dune conservation and hazard assessment.

**Abstract:**

Coastal sand dunes, shaped by aeolian and marine processes, are critical to natural ecosystems and human societies, making their morphological monitoring essential for effective conservation. However, large-scale, high-precision monitoring of topographic change remains a persistent challenge, a challenge that advanced sensing technologies can address. In this study, we propose an integrated, sensor-based approach using a UAV-mounted light detection and ranging (LiDAR) system, combined with a GNSS-RTK positioning unit and a novel ground control point (GCP) design to acquire high-resolution topographic data. Field surveys were conducted at four time points between October 2022 and February 2023 in the Tottori Sand Dunes, Japan. The digital elevation models (DEMs) derived from LiDAR point clouds achieved centimeter-level accuracy, enabling reliable detection of subtle topographic changes. Analysis of DEM differencing revealed that wind-driven sand deposition and erosion resulted in elevation changes of up to 0.4 m. These results validate the efficacy of the UAV-LiDAR sensor system for high-resolution, multitemporal monitoring of coastal sand dunes, highlighting its potential to advance the development of environmental sensing frameworks and support data-driven conservation strategies.

## 1. Introduction

Coastal sand dunes are highly dynamic landforms shaped by coupled aeolian–marine processes, and their short-term morphological changes have significant ecological and geo-environmental implications. However, coastal dunes can also generate wind-blown sand, which can damage human health and local infrastructure. Effective coastal dune management requires a good understanding of dune morphology changes at various spatial and temporal scales [1]. Although conventional ground surveys, such as total stations, provide techniques for monitoring topographic changes, they are labor-intensive, time-consuming, and often miss rapid events, e.g., [2,3]. Recent advances in remote sensing technologies, such as satellite imagery and optical and microwave sensors, have opened new possibilities for dune monitoring, notably through unmanned aerial vehicles (UAVs). In recent years, UAVs have also been increasingly applied for geological and environmental surveys, as well as for monitoring landslides and earthquake-related impacts, highlighting their versatility in environmental and geoscientific research, e.g., [4,5]. UAVs have been widely used for rapidly collecting data at high spatial–temporal resolutions. UAVs with structure-from-motion (SfM) photogrammetry have been demonstrated to be effective techniques for monitoring topographic changes in dunes, e.g., [6,7,8,9]. Nevertheless, light detection and ranging (LiDAR) sensors, which are active sensors that use laser pulses penetrating vegetation and delivering more geometrically stable ground elevations, acquire dense, three-dimensional point clouds with centimeter-level detail over large areas and short revisit intervals [10,11,12,13]. Focusing on dunes specifically, Pinton et al. [14] reported that ground elevations derived from UAV-LiDAR data were more accurate than those obtained from photogrammetry, even though a genetic algorithm was also applied.

Owing to the advantages of capturing low-texture surfaces such as sandy land, UAV-LiDAR has been increasingly applied to mapping coastal environments and geomorphic observations, e.g., [15,16,17]. For example, Lin et al. [9] reported that UAV-LiDAR can quantify rapid shoreline and dune changes with high fidelity. On dune coasts more broadly, reproducible LiDAR time-series workflows have been proposed to track beach and foredune morphology, underscoring the suitability of LiDAR for operational coastal monitoring [12]. Methodological approaches also include open-source toolchains that operate directly on LiDAR point clouds for shoreline and dune analysis, reflecting a maturing sensor-to-information pipeline for dynamic coasts [18]. However, the processing, calibration, and quality assessment of LiDAR point-cloud data accuracy still present challenges in geomorphological studies [19,20].

LiDAR data exhibit positioning errors due to their georeferencing accuracy, laser return density, and postprocessing procedures. GNSS-RTK (Global Navigation Satellite System–real-time kinematic) enables direct georeferencing, improving the accuracy of LiDAR and SfM data. Subsequent studies and technical assessments have compared UAV-LiDAR outputs with terrestrial laser scanning and photogrammetric systems, delineating the strengths, limitations, and calibration needs of compact UAV-LiDAR sensors [21,22,23]. These findings align with field practice, showing that GNSS-RTK integration and careful ground control improve absolute georeferencing, reduce vertical biases, and stabilize multiepoch differencing, which is crucial for dune-change detection [24]. Moreover, the quantity, distribution, and reliability of ground control points (GCPs) also influence the geometric quality of UAV-borne LiDAR products, e.g., [25].

While the impact of GCPs has been extensively studied, best-practice guidance for LiDAR-specific GCP design and deployment in low-texture sandy coastal environments remains limited [12,26]. Moreover, although UAV-LiDAR acquisition, RTK positioning, and DEM differencing are increasingly applied in coastal dune studies, multiepoch monitoring with compact UAV-LiDAR systems still suffers from epoch-dependent vertical bias and surface-thickness effects, which can obscure subtle geomorphic changes in dynamic sandy terrains. Operational procedures that explicitly ensure cross-epoch DEM consistency for repeated dune surveys are therefore still insufficiently addressed.

Against this background, we present an integrated sensing and processing approach that couples UAV-LiDAR (DJI Zenmuse L1) with GNSS-RTK and a LiDAR-specific GCP design to monitor short-term morphological dynamics at Tottori Sand Dunes, Japan. We conducted four surveys between October 2022 and February 2023 and developed a processing workflow to (i) evaluate point-cloud accuracy, (ii) calibrate vertical biases through fixed reference plots and GCPs, and (iii) compute multitemporal digital elevation model (DEM) differences and cross-sectional profiles to quantify erosion, deposition, and crestline migration. By explicitly addressing practical accuracy stabilization and providing a reproducible framework for long-term geomorphic observation in sandy coastal environments, this study advances the operational applicability of compact UAV-LiDAR for repeated dune monitoring. The results highlight the ability of UAV-LiDAR to resolve centimeter-scale elevation changes and meter-scale crest shifts over short time spans, thereby supporting science-based coastal conservation and hazard mitigation.

## 2. Materials and Methods

### 2.1. Study Area

The study was conducted in the central tourist area of the Tottori Sand Dunes in western Japan (Figure 1). The survey area covered approximately 1.34 km^2^ and was characterized by dynamic dune ridges and anthropogenic sand fences installed for conservation. UAV-LiDAR surveys were conducted on 12–13 October and 9 December 2022, and on 11 January and 27 February 2023. The total four survey campaigns were conducted with the same UAV-LiDAR system setup, including the LiDAR sensor, GNSS system and GCP configuration (see details in Section 2.2 and Section 2.3).

### 2.2. UAV-LiDAR System Setup

We used a DJI Matrice 300 RTK UAV platform equipped with a DJI Zenmuse L1 LiDAR payload (SZ DJI Technology Co., Ltd., Shenzhen, China; Table 1). L1 integrates a Livox Mid-40 LiDAR scanner (Livox Technology Company Limited, Shenzhen, China) and an RGB camera, producing true-color, georeferenced point clouds. The system was operated at a flight altitude of 100 m, speed of 8 m·s^−1^, and 70% front/side overlap. The LiDAR sensor has a beam divergence of approximately 0.28 mrad × 0.03 mrad (horizontal × vertical). At the flight altitude of 100 m above ground level, this corresponds to an elliptical laser footprint of approximately 2.8 cm × 0.3 cm on the ground. The system operates at a pulse repetition rate of up to 240 kHz, and under the applied flight parameters, the resulting average point density exceeded 200–300 points·m^−2^ over flat, non-vegetated surfaces. The UAV’s RTK module was linked with a GNSS base station to ensure centimeter-level georeferencing.

### 2.3. Ground Control Points (GCPs) and GNSS Survey

To further improve the absolute accuracy of UAV-LiDAR data, we designed LiDAR-optimized GCPs (Figure 2). Each GCP consisted of a 1 m × 1 m highly reflective white panel with a matte black cross pattern, mounted on a tripod 1 m above ground level. This design ensured high LiDAR-intensity contrast, facilitating reliable target recognition across multiple scan angles. Four control points and four independent checkpoints were distributed throughout the study area. The four control points were placed in the parking lots surrounding the sand dune, while the four checkpoints were located at the dune edge. Their coordinates were precisely measured with a TOPCON HiPer SR GNSS-RTK system (Topcon Corporation, Tokyo, Japan), which achieved a horizontal accuracy of ±0.01 m and a vertical accuracy of ±0.02 m.

### 2.4. Data Acquisition

LiDAR point clouds were collected via the DJI Pilot application with automated flight plans. Data were recorded in high-density mode, with raw LAS files including attributes such as RGB color, return intensity, and scan angle. The acquired data were first processed in DJI Terra software (version V4.1.0) to generate initial point clouds, which were projected to the JGD2011/Japan Plane Rectangular CS Zone V coordinate system.

### 2.5. Point Cloud Processing

The raw point clouds were further processed in TREND-POINT software (version 12; FUKUICOMPUTER, Inc., Fukui, Japan). A grid-based filtering method was applied to remove outliers and reduce point-cloud vertical noise. A ground surface extraction filter optimized for UAV-LiDAR data was first applied to remove vegetation such as trees and grasses. The filter was configured with a grid width of 2.0 m, a longitudinal window length of 5.0 m, and a vertical thickness threshold of 0.40 m. Then, point-cloud noise was reduced through a statistical noise filtering process, using a neighborhood distance of 0.25 m and a minimum neighbor count of 2. A near-neighbor point filtering step was subsequently applied with a search radius of 0.10 m, evaluated in three-dimensional space (XYZ), to remove isolated points while preserving continuous dune morphology. After that, ground-classified points were used to generate a triangulated irregular network (TIN) surface for each survey epoch. The minimum triangle edge length was set to 0.30 m to avoid degenerate triangles while maintaining micro-topographic detail. The TIN surfaces were rasterized to produce DEMs with a uniform spatial resolution of 0.50 m for all epochs. This grid resolution was selected based on point density and dune morphology to balance noise suppression and the representation of subtle elevation changes. All DEMs were generated using identical processing workflows and parameters to ensure cross-epoch consistency.

The GCPs and independent checkpoints were used to calibrate vertical offsets and ensure consistent alignment across survey epochs. GCPs located on stable and hard surfaces were employed to estimate epoch-specific vertical biases, whereas independent checkpoints were used for accuracy assessment. Additional quality control was performed by applying a vertical shift to each survey epoch based on fixed reference plots (parking lots, P1–P3, Figure 2), which were assumed to be temporally invariant. For each epoch, the mean elevation difference between LiDAR-derived elevations and GNSS-RTK–measured reference heights within these plots was calculated and applied as a uniform vertical offset to the point cloud. This procedure effectively reduced systematic vertical bias and minimized the influence of point-cloud thickness, resulting in centimeter-level vertical residuals at the independent checkpoints.

### 2.6. DEM Differencing and Change Detection

Following the point cloud processing, vertical bias calibration, and quality control described above, high-accuracy UAV-LiDAR point clouds with consistent vertical alignment across survey epochs were obtained. These calibrated point clouds were then used to generate digital elevation models (DEMs) and to quantify topographic changes through DEM differencing.

Multitemporal DEM differencing was performed to detect elevation changes associated with sand deposition and erosion. Terrain profiles were extracted along three transect lines to track crestline migration and dune height variations (Figure 2). The transect lines L1, L2, and L3 were perpendicular to the first, second, and third rows of dunes, respectively, and were oriented approximately from northwest to southeast [27].

In addition, volumetric changes were computed for areas surrounding the sand fences. Volumetric changes between survey epochs were estimated using surface differencing implemented through the volume calculation module in TREND-POINT. Elevation differences were computed as DEM(t_2_) − DEM(t_1_), where t_1_ and t_2_ denote the earlier and later survey epochs, respectively. Positive values indicate deposition, and negative values indicate erosion. Volume calculations were performed within a manually defined dune polygon, which was consistently applied to all four survey epochs to ensure that identical terrain areas were compared.

Wind speed and direction were measured with an anemometer (CYG-5108, R. M. Young Co., Traverse City, MI, USA), which was installed and fixed at the center of the sand dune by the Department of the Environment and Consumer Affairs, Tottori Prefecture (Figure 2). The observed wind data were integrated to interpret the relationship between the observed topographic changes and prevailing wind conditions.

The overall workflow is shown in Figure 3.

## 3. Results

### 3.1. Vertical Calibration and Alignment Assessment

The accuracy of the generated point cloud was first evaluated using direct RTK-based georeferencing by computing the mean difference between the original point cloud and GNSS-measured check points. Horizontal (XY) differences were approximately 0.01 m, whereas vertical (Z) differences ranged from 0.17 m to 0.42 m (Figure 4) for the four checkpoints. These results indicate that although the point-cloud coordinates achieved centimeter-level horizontal accuracy, the vertical accuracy was degraded by the relatively large thickness of the DJI L1 LiDAR point clouds (ranging from 0.068 to 0.19 cm), resulting in substantial elevation uncertainty. Therefore, we applied additional processing to the point-cloud data.

Three parking lots (P1–P3), each covering an area of approximately 15 m × 10 m with fixed positioning, were selected as calibration areas. Measurements in the P4 area were excluded as calibration data because the ground elevation at this parking lot was not stable, as the surface was not paved with concrete. We then determined the elevation used for the shift in the vertical axis on the basis of a comparison of the elevation differences between the two observation periods. Although the elevation differences were similar (approximately 0.05 m) at P1 and P2, the elevation difference at P3 decreased from 0.1 m to 0.05 cm (Figure 5). We suggest that the vertical-shift-based calibration using stable hard-surface reference areas effectively reduces systematic vertical bias and improves the quality and cross-epoch consistency of LiDAR-derived point clouds.

The effectiveness of the GCP-based vertical calibration is summarized in Table 2, which reports the mean bias, standard deviation, and RMSE of vertical residuals at GCP locations after calibration for each survey epoch. Following calibration, mean vertical biases were small, ranging from −0.027 m to +0.027 m across all surveys, indicating that epoch-dependent systematic offsets were effectively removed. The corresponding RMSE values ranged from 0.012 m to 0.045 m, demonstrating centimeter-level vertical alignment accuracy after calibration. Results indicate that the GCP-based calibration substantially improves cross-epoch vertical consistency of the UAV-LiDAR-derived DEMs, providing a robust basis for subsequent DEM differencing and change-detection analyses.

### 3.2. Sand Deposition and Erosion

To monitor sand deposition and erosion in the sand dune areas, we examined the wind distributions and elevation differences in the point clouds between October and December 2022, December 2022 and January 2023, and January and February 2023 (Figure 6). In addition to spatial elevation-change maps, histograms of elevation differences were used to characterize the statistical distribution of deposition and erosion magnitudes for each period. Sand movement tended to occur with increasing frequency of strong winds for speeds exceeding 10 m s^−1^ and wind directions from northwest to northeast during the observation period. In particular, deposition and erosion characterized by elevation changes greater than 0.5 m occurred mainly in the second row of sand dunes.

From October to December 2022, the predominant wind direction was from southern elevation changes (Figure 6a). Sand was transported northward of the second row of dunes by winds, after which deposition started. Approximately 5.5 × 10^4^ m^3^ of sand deposition occurred, mainly northwest of the dune (Figure 7). Between December 2022 and January 2023, winds with speeds greater than 10 m s^−1^ blew from the northwest, resulting in the opposite direction of sand movement (Figure 6b). Affected by strong winds, approximately 6.8× 10^4^ m^3^ of sand deposition occurred (Figure 7), mostly southeast of the second row of the sand dune and some around the third row of the sand dune. In the eastern part of our study area, where sand fences were constructed, wind-blown sand was trapped by the fences, resulting in sand deposition. Strong winds from the northwest to the northeast increased in January and February, resulting in a wide distribution of sand deposition (Figure 5c). Approximately 3.5 × 10^4^ m^3^ of sand deposition during this period continuously occurred across the study area (Figure 7). Sand deposition southeast of the second row of sand dunes and around the third row of sand dunes became more severe.

To detect topographic changes in the sand dune areas, three terrain profiles depicting the elevations of the cross-sections were examined on the basis of the differences in the point clouds between October and December 2022, December 2022 and January 2023, and January and February 2023. Elevation changes were detected in the sand dune areas at L2. The top elevation of the sand dune in L2 was 42.4 m in October and then gradually decreased. From January to February, the top elevation decreased by approximately 0.4 m (Figure 8b).

## 4. Discussion

The results confirm the effectiveness of integrating UAV-LiDAR with GNSS-RTK and an optimized GCP design for detecting subtle morphological changes in coastal dune environments. The centimeter-level DEM accuracy aligns with findings from previous UAV-LiDAR validation studies. For example, Štroner et al. [20] reported centimeter-scale accuracies for the DJI Zenmuse L1 under controlled conditions, whereas Pinton et al. [13] demonstrated that UAV-LiDAR can resolve dune-crest shifts and elevation changes with high reliability. Our results fall within the same accuracy range and further highlight the benefits of applying a LiDAR-specific GCP design to reduce vertical biases across multitemporal surveys.

Methodologically, the design and deployment of GCPs tailored to LiDAR-intensity values represent an important innovation. While GCP optimization, including number and spatial arrangement, has been widely studied in UAV photogrammetry, e.g., [28,29,30], few studies have assessed its role in UAV-LiDAR workflows. Unlike SfM approaches in photogrammetry, UAV-LiDAR point clouds are directly georeferenced using onboard GNSS/IMU measurements, and their geometric rigidity does not rely on dense internal GCP configurations [31,32,33]. Accordingly, GCPs in this study were primarily used to assess and correct residual systematic errors, particularly epoch-dependent vertical bias. Although densely distributed GCPs usually reduce systematic errors, using high-accuracy GNSS-RTK for georeferencing enabled centimeter-scale accuracy with a limited number of GCPs across the survey area. In addition, installation of artificial markers or ground disturbance within the dune interior is strictly prohibited because the Tottori Sand Dunes are designated as a “Special Protection Zone of the San’in Kaigan National Park”. Consequently, we employed the peripheral GCP configuration in this study. Our findings suggest that careful GCP placement, combined with postprocessing of LiDAR point clouds, substantially improves DEM consistency, especially in low-texture sandy environments. However, point-cloud thickness can introduce significant vertical uncertainty, causing errors when differencing DEMs from different time periods. Therefore, we propose a method for evaluating and calibrating accuracy on the basis of the DEM difference between two surveys with GCPs on hard surfaces and parking lots in this study. Focusing on hard-surface regions has been suggested to estimate a detection threshold, especially for fluvial environments, e.g., [34,35,36], whereas discussion of this methodology for coastal dunes is limited. In addition, this study not only validated but also further calibrated and improved the vertical accuracy and consistency of the DEMs. This underscores the need for sensor-specific calibration strategies in UAV-based topographic monitoring. Nevertheless, applications requiring centimeter-scale accuracy or operating under degraded GNSS/IMU conditions may require additional stable reference surfaces outside protected areas or alternative control strategies. This limitation should be considered when transferring the proposed workflow to other sites.

In this study, surveys were conducted at an altitude of 100 m, where the laser footprint diameter is approximately 5–6 cm, assuming a beam divergence of ~0.52 mrad. However, vertical discrepancies observed at GCPs (17–42 cm) primarily reflect point-cloud thickness rather than instrument ranging precision [37]. This thickness arises from a combination of factors, including non-repetitive scan geometry of the Livox Mid-40 sensor, oblique incidence angles, variable pulse density, and heterogeneous laser–sand interactions. Unlike conventional repetitive scanning LiDAR systems, the Livox sensor employs a rosette-like, non-repeating scan pattern, resulting in spatially variable sampling geometry between epochs, e.g., [38]. Over low-texture sandy surfaces, this leads to vertical dispersion of returns within each footprint and amplifies apparent elevation variability [14,20]. Consequently, the observed discrepancies exceed nominal brochure specifications but remain consistent with previous UAV-LiDAR evaluations over natural, low-reflectance terrain. To mitigate these effects, we applied surface-based averaging, vertical bias calibration using stable reference plots, and DEM-level uncertainty filtering, which reduced effective vertical uncertainty to the centimeter level for DEM differencing analyses.

On the basis of the derived high-accuracy DEM datasets, the observed dune elevation changes of up to 0.4 m and crestline shifts of approximately 6 m are consistent with studies linking wind-driven sand transport to rapid dune dynamics. Lin et al. [8] and Le Mauff et al. [9] emphasized that UAV-LiDAR is well-suited for capturing short-term geomorphic responses to wind events and storms in coastal systems. By coupling UAV-LiDAR surveys with wind records, our study reinforces the role of high-resolution sensing in linking environmental forcing to geomorphic change. This integration underscores the potential of UAV-LiDAR not only for dune monitoring but also for broader applications in coastal hazard assessment and conservation planning. However, such applications would require environment-specific validation, as surface properties and sampling conditions differ substantially among landform types.

Quantifying uncertainty is a critical requirement for DEM-of-difference-based geomorphic change detection, particularly in low-relief sandy environments where subtle elevation changes may be comparable to measurement noise. Previous studies have demonstrated that uncertainty-aware frameworks, such as minimum detectable change (MDC) or level-of-detection (LoD) approaches, are essential for robust interpretation of surface change [36]. However, a full algorithmic implementation of uncertainty-propagated MDC was not feasible within the current processing environment. To avoid overstating the quantitative reliability of the detected changes, the geomorphic results presented in this study are interpreted primarily in a relative and pattern-based sense, rather than as strictly precise magnitudes. Future work will incorporate uncertainty-aware DEM differencing frameworks that explicitly estimate spatially variable DEM error and minimum detectable change thresholds, using open-source or custom processing pipelines that allow full uncertainty propagation. Such approaches will enable more rigorous quantification of volumetric change and crestline migration uncertainty and will further strengthen the interpretation of UAV-LiDAR-based dune monitoring results.

Long-term monitoring campaigns utilizing UAV-LiDAR with an array of sensors (e.g., multispectral or hyperspectral imaging) in order to jointly assess geomorphological and ecological dynamics are an important avenue of future research. The continued development of computer-aided processing methods for point clouds, sensor fusion, and machine-learning applications could further enhance the scalability of UAV-LiDAR monitoring. Finally, collaborative coastal observatories combining UAV-LiDAR data with in situ environmental measures may shed more light on the complex interactions of climate forcing, sediment transport, and dune-system resilience.

## 5. Conclusions

This study demonstrates the effectiveness of an integrated UAV-LiDAR sensor system, combined with GNSS-RTK and an optimized GCP design, for monitoring the short-term morphological dynamics of coastal sand dunes. By conducting four surveys between October 2022 and February 2023 at the Tottori Sand Dunes, Japan, we generated centimeter-level DEMs that enabled the reliable detection of subtle dune changes. Revealed elevation variations of up to 0.4 m, closely linked to prevailing wind conditions during the observation periods.

Methodologically, this work highlights the importance of GCP configuration and vertical calibration in enhancing UAV-LiDAR point-cloud accuracy. Our proposed GCP design and postprocessing workflow reduced vertical biases and improved the consistency of DEM differencing, thereby expanding the potential of UAV-LiDAR for repeat geomorphic monitoring. These findings contribute to the growing body of knowledge on sensor calibration and integration for high-resolution environmental sensing.

Practically, this study confirms that UAV-LiDAR is a powerful tool for coastal dune conservation and hazard management, offering an efficient and scalable alternative to labor-intensive ground surveys. In addition to coastal dunes, the methodological framework presented here can be extended to other dynamic landscapes, such as river channels, glacier surfaces, and landslide-prone areas—where centimeter-scale elevation changes are critical for detection. Future work should integrate UAV-LiDAR with additional sensors (e.g., multispectral or hyperspectral imaging) to enable multiparameter monitoring of coastal systems and assess the long-term value of such sensor networks for adaptive coastal management.

## Figures and Tables

**Figure 1 sensors-26-00302-f001:**
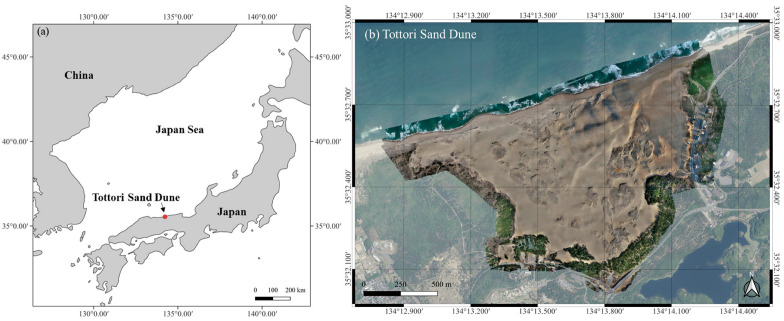
(**a**) Location of the study area in the Tottori Sand Dunes, western Japan, and (**b**) the extent of the UAV-LiDAR survey zone.

**Figure 2 sensors-26-00302-f002:**
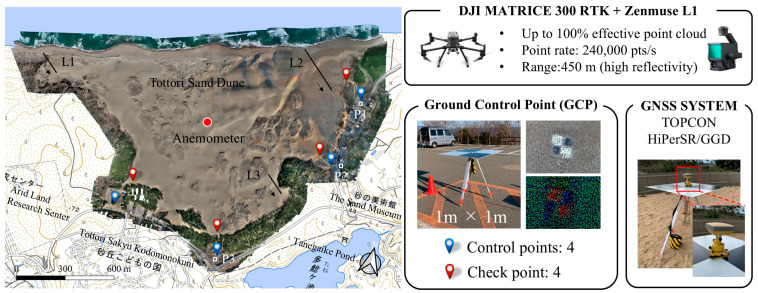
The UAV-LiDAR survey setup includes a drone platform (DJI Matrice 300 RTK), a LiDAR sensor (Zenmuse L1), a GNSS system (TOPCON HiPer SR), and custom-designed ground control points (GCPs). Locations of the four control points (red markers), four checkpoints (blue markers), three transect lines (L1–L3), anemometer (red dots), and parking lots (P1–P3) used for vertical calibration. Arrows along the transect lines indicate cardinal directions.

**Figure 3 sensors-26-00302-f003:**
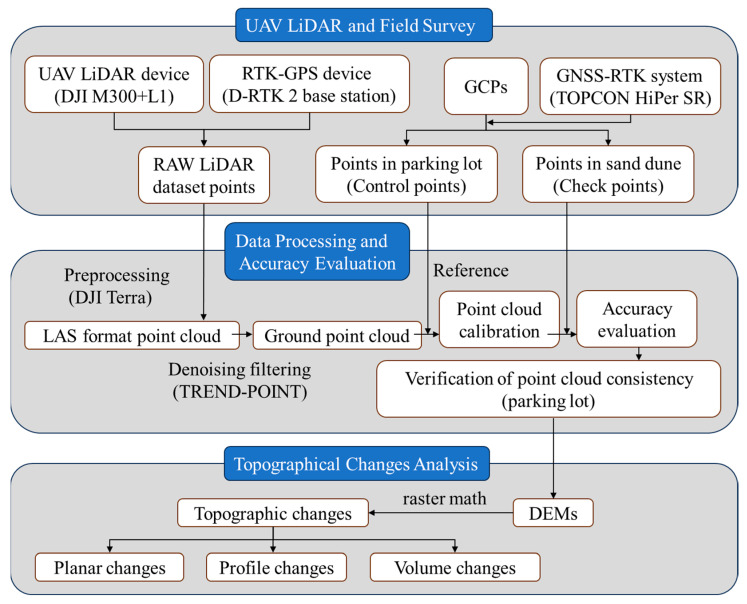
Workflow of the UAV-LiDAR sensing framework used in this study. The process includes UAV-LiDAR data acquisition, GNSS-RTK and ground control point (GCP) surveys, point cloud processing, DEM generation, multitemporal DEM differencing, and change analysis (crestline migration and volumetric erosion/deposition).

**Figure 4 sensors-26-00302-f004:**
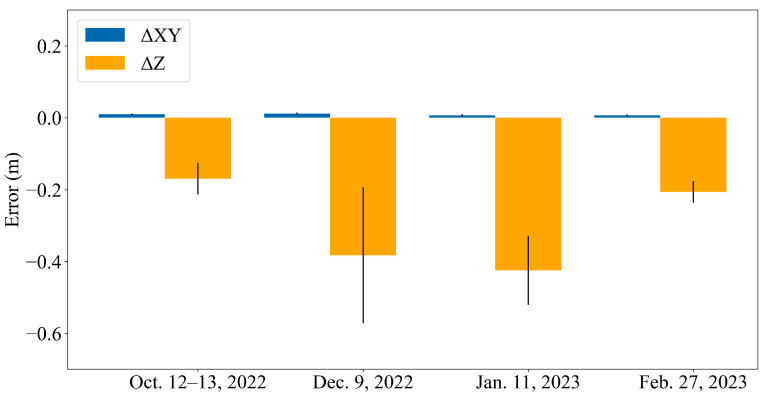
Horizontal and vertical accuracy assessment of ground control points measured with GNSS-RTK during UAV-LiDAR surveys.

**Figure 5 sensors-26-00302-f005:**
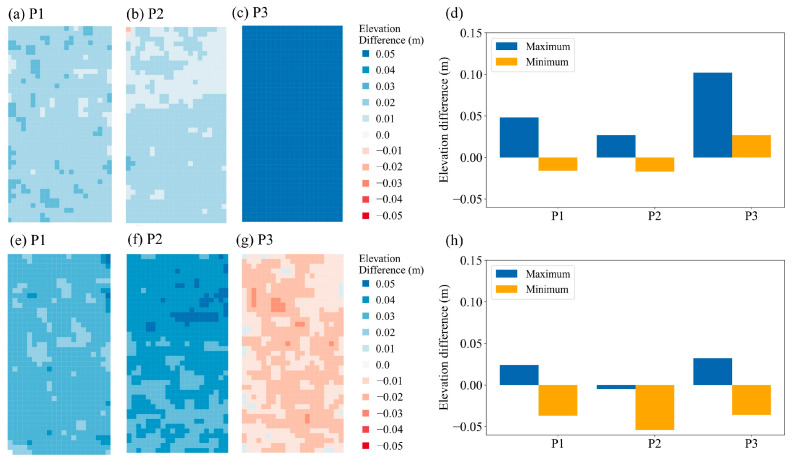
Elevation differences between December 2022 and January 2023 at three reference plots: (**a**–**d**) without GCP calibration and (**e**–**h**) with GCP calibration.

**Figure 6 sensors-26-00302-f006:**
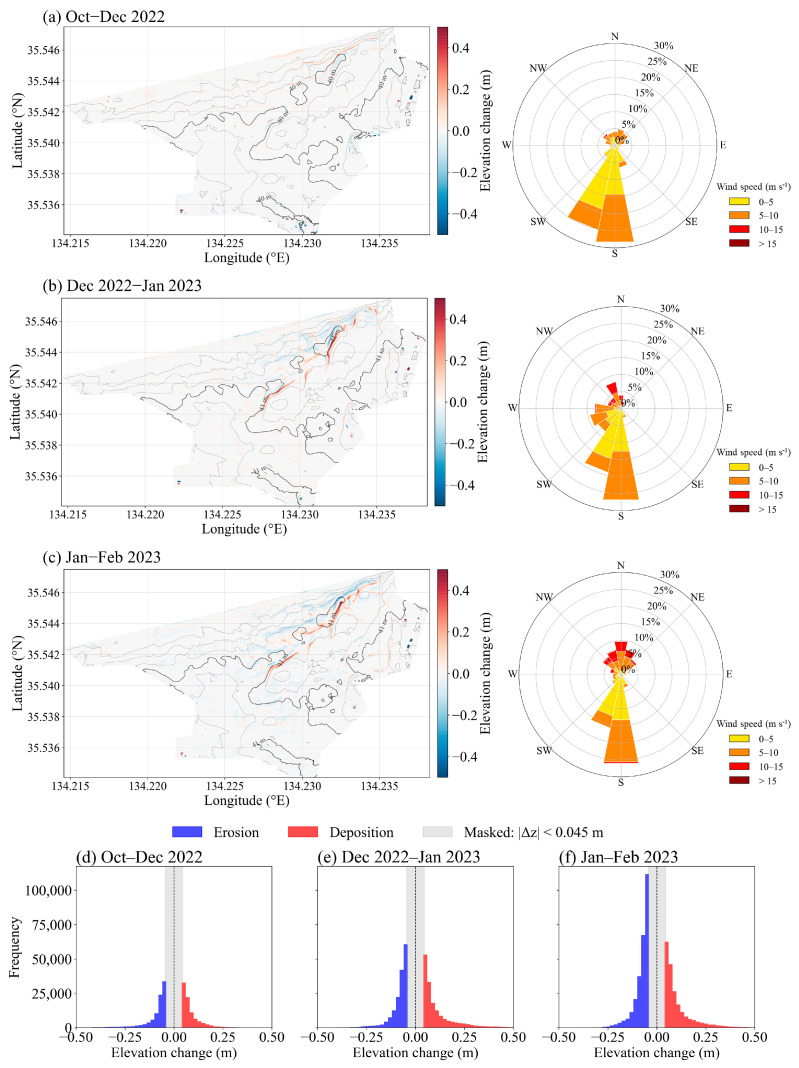
Spatial distribution of elevation changes in the study area, with prevailing wind directions indicated, for the periods: (**a**) October–December 2022, (**b**) December 2022–January 2023, and (**c**) January–February 2023. Positive values represent sand deposition, whereas negative values indicate erosion. Frequency distributions of elevation changes derived from DEM differencing for the same periods: (**d**) October–December 2022, (**e**) December 2022–January 2023, and (**f**) January–February 2023. Elevation changes with absolute values smaller than 0.045 m are masked (gray) to suppress minor variations likely associated with measurement noise. The dashed vertical line denotes zero elevation change.

**Figure 7 sensors-26-00302-f007:**
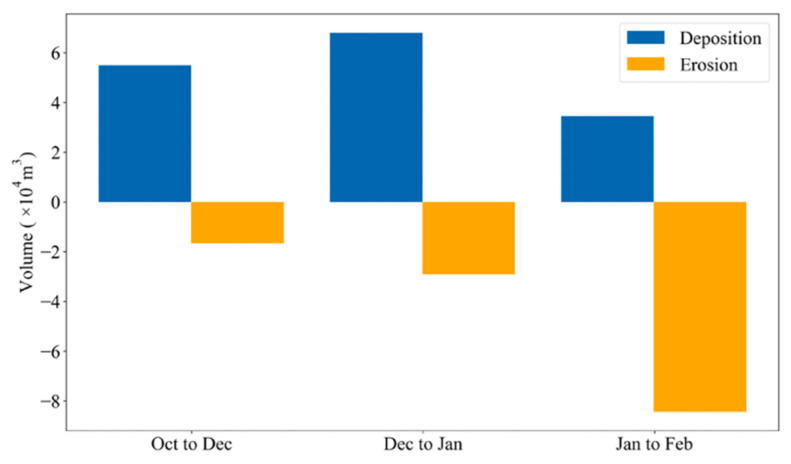
Volumetric changes in sand deposition and erosion around two sand fence areas for the three survey intervals: October–December 2022, December 2022–January 2023, and January–February 2023.

**Figure 8 sensors-26-00302-f008:**
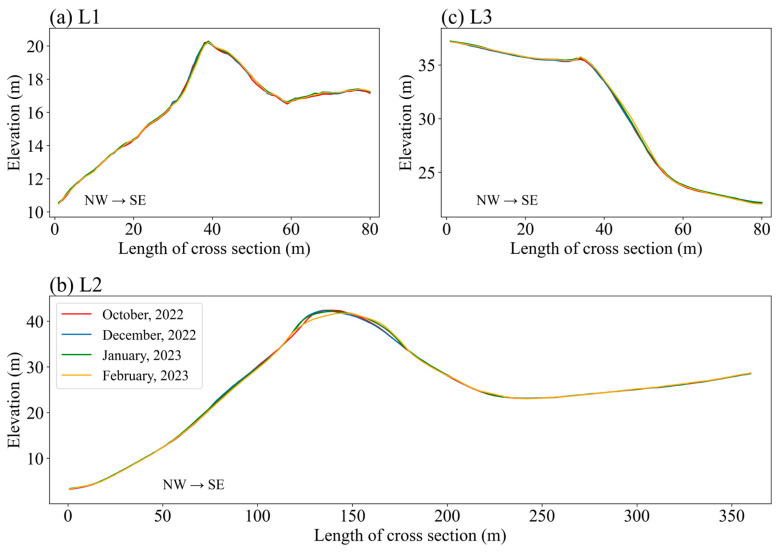
Elevation profiles along three transect lines (**a**) L1, (**b**) L2, and (**c**) L3 in October (red), December (blue), January (green), and February (yellow), showing crestline migration and dune height changes.

**Table 1 sensors-26-00302-t001:** Technical specifications of the UAV platform, LiDAR sensor, GNSS system, and ground control points (GCPs) used in this study.

Component	Model/Description	Key Specifications
UAV platform	DJI Matrice 300 RTK	Maximum flight time: ~55 min; Flight speed: 8 m·s^−1^; Flight altitude: 100 m; Front/side overlap: 70%
LiDAR sensor	DJI Zenmuse L1	LiDAR type: Livox Mid-40; Wavelength: 905 nm; Field of view: 70.4° (circular); Max range: 450 m (reflectivity dependent); Point rate: up to 240,000 pts·s^−1^; Integrated 1-inch CMOS RGB camera (20 MP) for true-color point clouds
GNSS system	TOPCON HiPer SR (GNSS-RTK)	Positioning accuracy: Horizontal ±0.01 m; Vertical ±0.02 m; Dual-frequency RTK
Ground control points (GCPs)	Custom-designed targets	Size: 1.0 m × 1.0 m; Material: High-reflective white foil board with black matte cross pattern; Mounted on tripods at 1 m height above ground; Precisely surveyed by GNSS-RTK

**Table 2 sensors-26-00302-t002:** GCP-based vertical accuracy after calibration (GCPs).

Date	n (GCPs)	Mean Bias (m)	Std (m)	RMSE (m)
12–13 October 2022	4	+0.027	0.019	0.031
9 December 2022	4	+0.011	0.007	0.012
11 January 2023	4	−0.017	0.017	0.023
27 February 2022 *	3	−0.027	0.043	0.045

* One outlier GCP (ΔZ = −0.146 m) was excluded.

## Data Availability

The UAV-LiDAR point clouds, DEMs, and GNSS-RTK ground control data generated in this study are available upon request from the corresponding author. The data are not publicly available due to institutional restrictions but can be provided for academic research purposes.

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
