# Peer review of "Detection and Monitoring of Topography Changes at the Tottori Sand Dune Using UAV-LiDAR"

_sensors, 2026, doi:10.3390/s26010302_

Round 1

Reviewer 1 Report

Comments and Suggestions for Authors

Dear Editor and Authors,

I have reviewed the manuscript titled “Detection and Monitoring of Topography Changes at the Tottori Sand Dune using UAV-LiDAR”. The article is very interesting and highly relevant, particularly in terms of the methods employed.

However, before it can be considered for publication in Sensors, I believe it requires substantial improvements. I have assigned a major revision and provided detailed comments directly in the manuscript PDF.

I encourage the authors to carefully consider these comments and address them thoroughly. Implementing these suggestions will undoubtedly enhance the article's quality and clarity.

Thank you for your attention, and I look forward to reviewing the revised manuscript.

Reviewer 2 Report

Comments and Suggestions for Authors

Dear authors, I have reviewed your manuscript “Detection and Monitoring of Topography Changes at the Tottori Sand Dune using UAV-LiDAR” (sensors-3989404). I have enjoyed reading it and I think that the paper includes a very interesting practical work on dune dynamics by means of LiDAR UAV. For me, it has been quite interesting how geomatics techniques are applied to a practical field problem like the presented in this paper. The inclusion of wind data and the relation with the dune dynamics is also very interesting.

My main concern is about the validation of the LiDAR point cloud at the plots. It is not clear for me how you have made such validation. May be you can explain better the process.

Moreover, you have presented the target design as a novelty, but such targets are usually employed on both laser and photogrammetry surveys (you should comment that).

Another question I would like to comment is about the DSM data processing. Have you tried to detect (automatically) and compare the dune crests between different epochs? Your analysis is based on volume changes between different epochs and profile comparisons in the transects, but you could try to explore the analysis of the evolution of different features (like the crests) through a broader DSM analysis.

Finally, since you have also captured the images with the Zenmuse sensor, have you compute the DSMs derived from photogrammetry? May you could establish differences between both methodologies.

Next you can find some specific comments to your work. I hope you can find them useful to improve the paper.

Best regards

LINES/PAGES

Observations

Table 1

With respect to the flight planning, how many flight lines were performed to cover the whole area?

122

This GCP design (in fact it is a checkerboard) is not a novelty and they have been used for example with FARO scans. Also in photogrammetry are widely used (for example Metashape can automatically measure non coded targets in images based on this design)

126-…

In my opinion, the distribution of both GCP and Check points can be improved since all eight points are located on the south and east part of the periphery of the area. There isn’t any point in the center and west and north part of the area. Moreover, chk points are located just near the CGP. The should be located far from their influence (for example a chk had been interesting in the center area).

May be you can not set any point in the center of the area because you cannot access to that area for environmental protection reasons or to avoid any disturbance in the dune surfaces? If you can access to that areas points should have been located there

145

Could you give some information about the amount of noise (in cm) present in the raw point cloud captured in the sand with the DJI Zenmuse L1?

167-172

I don’t think that this is the accuracy of the point cloud. First the residuals at CGP are not significant. They are useful to set the reference system and to remove systematic errors. Real errors can be identified at the check points, but maybe the residuals at cm level in the chk points are not exactly the noise at the sand dune surface, since chk/GCP points can be measured automatically in the checkerboards.

173

Does “thickness” mean noise? Can you give the amount of such thickness.

175-183

How do you measure the height differences? Do you compare the difference at computed plot surfaces between two epochs? Or do you compare the distances between two point clouds in two epochs. If you use two different epochs, what epochs are?  Please could you explain more clearly how do you improve or calibrate the height of the point clouds? I don’t fully understand the process and exactly what you are comparing

195-196

“In particular, deposition and erosion at depths greater than 0.5 m occurred 195 mainly in the second row of sand dunes. ¨ May be I’m not used to the terminology in analysing dune dynamics, but instead of “at depths” I’d used height differences or similar. But maybe in this context  is correct “at depths”.

201-214

If you have close photographs of sand trapped by the fences such images would very illustrative for non-experts in dune dynamics (like me…)

237-238

As mention before the target design for laser measurements is not a novelty.

242

More than across the area, the point were only distributes in some parts of the periphery.

Reviewer 3 Report

Comments and Suggestions for Authors

THis is a review report of the manuscript entitled "Detection and Monitoring of Topography Changes at the Tot-2 tori Sand Dune using UAV-LiDAR"

This manuscript addresses UAV-LiDAR-based detection of coastal dune dynamics, a topic that falls within the scope of Sensors. However, in its current form, the manuscript is not suitable for publication and requires major revision before it can be reconsidered.

The study suffers from limited novelty, insufficient methodological transparency, weak accuracy/uncertainty analysis, and descriptive-level interpretation that does not meet Sensors standards for rigorous sensor performance evaluation.

Major Concerns
1. Limited novelty / incremental contribution

The manuscript does not clearly advance the state of the art in UAV-LiDAR–based dune monitoring. Most components of the workflow (UAV-LiDAR acquisition, GNSS-RTK integration, DEM differencing) already appear extensively in the literature cited by the authors themselves.
A clearer articulation of what is technically or methodologically new is required.

Specific questions that must be addressed:

Compared to Pinton et al. (2023), Elsner et al. (2018), and other cited works, what exactly is novel?

How does the proposed LiDAR-specific GCP design differ from existing designs?

In what way does this study advance or improve upon prior work at the Tottori Sand Dunes?

2. Insufficient methodological description (limits reproducibility)

Several core methodological steps are under-explained, preventing reproducibility and making it difficult to assess scientific rigor.

DEM creation

No information on grid resolution

No interpolation method (TIN, IDW, kriging, etc.)

No explanation of how outliers, voids, or noise were handled

Ground filtering

Only a vague “grid-based filter” is described

No algorithm name, no filtering parameters, no justification

No assessment of filtering performance over low-texture sand surfaces

Volumetric change estimation

No details on buffer width, masking logic, separation of erosion/deposition, or thresholding

Without transparent methodology, volumetric estimates cannot be validated

This level of detail is insufficient for publication in Sensors, which emphasizes rigorous reproducibility of sensor-derived workflows.

3. Accuracy and uncertainty analysis is inadequate

This is the most serious flaw for a study claiming centimeter-level precision.

3.1. Vertical accuracy

The manuscript reports 0.17–0.42 m vertical discrepancies at GCPs

Authors then correct using “parking lot elevation shifts,” but:

No RMSE, STD, confidence intervals, or pre/post calibration metrics are provided

No error propagation is shown

It is unclear how this correction remains valid across the dune field

What happens in areas without any stable features?

3.2. DEM differencing uncertainty

Change detection is reported without a minimum detectable change (MDC)

Wheaton et al. (2010) is cited but not applied

Crestline shift (~6 m) and dune lowering (~0.4 m) lack uncertainty bounds

The authors must explicitly quantify uncertainty and show:

Local DEM error

Global DEM error

DEM-of-difference (DoD) threshold

Propagated uncertainty in volumetrics and crestline displacement

Without this, geomorphic changes are not scientifically reliable.

4. GCP distribution is suboptimal and insufficiently justified

This is a major structural weakness in the data acquisition design.

Only four control-point GCPs and four checkpoints were deployed over ~1.34 km²

All control GCPs are located in parking lots around the dune, and all checkpoints at the dune edges

No GCPs or checkpoints exist inside the dunes, where elevation variation is largest

The central dune area is therefore geometrically unconstrained

In such a highly dynamic, low-texture environment, this configuration cannot ensure:

Block-level stability

Prevention of warping/tilt

Robust vertical alignment

The manuscript should either:

(i) Provide a strong justification for why this sparse peripheral GCP layout is adequate,
or
(ii) Clearly acknowledge this as a limitation and discuss its implications for DEM reliability.

Ideally, a sensitivity analysis comparing:

alternative GCP layouts

additional internal GCPs

simulated distributions

would be needed to support the claim that the proposed “LiDAR-optimized GCP design” is generalizable and not site-specific.

5. Analysis is largely descriptive and lacks quantitative depth

The manuscript presents maps and profiles but does not include:

Statistical quantification of spatial change patterns

Correlation of wind forcing with measured deposition/erosion

Regression analysis or threshold analysis for wind speed

Time-series analysis of crestline movement

Comparative evaluation with other sensors (SfM, TLS, airborne LiDAR)

Thus, the study remains descriptive and does not reach the analytical depth expected for Sensors, particularly regarding sensor performance assessment.

6. Limited generalizability and narrow temporal scope

Study area: one well-studied dune system

Time span: four surveys over a few months

Claims of applicability to rivers, glaciers, and landslides are unsupported, as no such surface types are tested or even discussed in detail

Statements about broad applicability must be supported by evidence or removed.

7. Presentation and clarity issues

Although individually minor, together they weaken scientific rigor.

Several figures (e.g., DEM differencing maps) have poor contrast and small labels

Highlights and Abstract repeatedly claim “centimeter-level accuracy” without distinguishing between:

Local relative accuracy after calibration

Global absolute accuracy (which is not centimeter-level)

Line 180–181: 0.05 cm appears to be a unit error (likely 0.05 m)

Wind speed units need correction (e.g., m s⁻¹ consistently)

Parking lot photos or orthophotos should be added to confirm surface stability

The manuscript requires thorough proofreading.

8. DJI L1 specification mismatch

The DJI Zenmuse L1 brochure provides:

±10 cm horizontal accuracy @ 50 m

±5 cm vertical accuracy @ 50 m

However, the authors flew at 100 m, where expected vertical accuracy is ~10 cm.

The manuscript reports 17–42 cm vertical discrepancies, which is:

3–8× worse than expected performance

Not adequately explained

Not quantified

Not accompanied by an error budget

Not contextualized with laser beam divergence, footprint size, or scan geometry

The paper must supply:

Beam divergence

Footprint size at 100 m

Pulse density

Sampling geometry explanation

Discussion of the L1’s non-repetitive Livox scan pattern

Without this, vertical errors remain unexplained.

Conclusion

Due to the issues above—particularly the insufficient uncertainty analysis, inadequate GCP distribution, and lack of methodological transparency—the manuscript is not suitable for publication in its current state.

A substantially revised and expanded version, with rigorous quantification of uncertainty, stronger methodological justification, clearer novelty, and improved presentation, would be required for reconsideration.

Round 2

Reviewer 1 Report

Comments and Suggestions for Authors

The manuscript entitled “Detection and Monitoring of Topography Changes at the Tottori Sand Dune using UAV-LiDAR” presents a solid and well-structured study, with a clear methodology and relevant results in the field of topography monitoring using UAV-LiDAR.

The authors have satisfactorily addressed all comments and suggestions provided during the review process, further improving the quality of the manuscript. The content is original, scientifically sound, and of interest to the readership of Sensors. No further major changes are required; the manuscript can be accepted for publication.

Reviewer 2 Report

Comments and Suggestions for Authors

Dear authors, thank you for your efforts in answering all my questions and suggestions. In my opinion the paper has been improved and now some sections are clearer. 

Finally, as we approach 2026, I wish you a happy and prosperous New Year.

Reviewer 3 Report

Comments and Suggestions for Authors

My previous comments are sufficiently replied.